# Plasma-Polymerized Aniline–Diphenylamine Thin Film Semiconductors

Claudia Nastase [1], Gabriel Prodan [2] and Florin Nastase [1],*

[1] National Institute for Research and Development in Microtechnologies, 126A Erou Iancu Nicolae Street, Voluntari, 077190 Ilfov, Romania

[2] Department of Physics, Faculty of Applied Sciences and Engineering, Ovidius University of Constanta, Bvd. Mamaia, No 124, 900527 Constanta, Romania

* Correspondence: florin.nastase@imt.ro

**Abstract:** Semiconducting polymer thin films were grown by a DC plasma-polymerized technique using a mixture of aniline–diphenylamine as a precursor. FT-IR spectra were taken in order to analyze the structural properties of the resulting polymers. From morphological and structural studies performed by transmission electron microscope (TEM) and X-ray diffraction, an organized structure in plasma polymer thin films was distinguished. I–V characteristics in an asymmetric electrode configuration were studied to determine the conduction mechanism. It was found that the conduction mechanism controlled by SCLC is dominant in plasma-polymerized aniline–diphenylamine (PPAni-PDPA) thin films.

**Keywords:** semiconducting polymers; plasma polymerization; electrical properties; thin film

## 1. Introduction

Polyaniline (PAni) is one of the most studied polymers from the conjugated polymer class due to its ease of synthesis by chemical and electrochemical methods [1,2], good environmental stability [1], and electrochromism [3,4]. The remarkable properties of PAni make this polymer of great interest in applications such as light-emitting diodes (LEDs), rechargeable batteries [5,6], supercapacitors [7,8], solar cells [9,10], and, in the last few years, fuel cells [11,12].

Intense research into N-substituted aniline derivatives [13] is necessary due to the increasing demand for conducting polymers for application as electrode materials, microelectronics, and electrochromic materials [14]. PAni has suitable properties such as increased processability [15] and electrochromism [16].

N-aril-substituted polymers represent a class of conducting polymers that are intermediately between polyaniline and polyphenylene [16]. Polydiphenylamina (PDPA), an N-aril-substituted aniline polymer, presents many properties that are not comparable with other polymers of N-substituted aniline [17,18].

Different polymers of N-substituted aniline have been obtained in aqueous media [16,17] or nonaqueous media, such as acetonitrile media [18], by electrochemical methods. Cyclic voltammetry and chemical methods [19,20] have been used for the preparation of stable and adherent polydiphenylamine at the work electrode surface. Additionally, it was discovered that diphenylamine polymerizes in p-toluene sulphonic acid [21] as well as methane sulfonic acid-doped polydiphenylamine [20].

In recent years, polyaniline/polydiphenylamine copolymers have been investigated because of their specific characteristics and potential industrial applications as supercapacitors and electrocatalysts in fuel cell devices [22–24].

Poly(diphenylamine-co-aniline) copolymers with a high content of anilinium dodecyl sulfate, synthesized by oxidative polymerization, have characteristics that suggest a feasible application in electrodes in supercapacitor designs [22]. Fluorine-doped tin

oxide/polydiphenylamine–polyaniline/phosphotungsticacid nanohybrid-modified electrodes have been fabricated as electrode material in supercapacitor device applications [23]. Polyaniline–polydiphenylamine copolymer nanocomposites with Pd nanoparticles have been successfully synthesized as electrocatalysts for methanol and ethanol oxidation reactions [24].

Although chemical and electrochemical polymerization is still used for the deposition of polyaniline and thin films on metal electrodes, the plasma polymerization technique is increasingly being used as an alternative for obtaining thin polymeric films [25–27]. The plasma polymerization technique presents some advantages over conventional chemical and electrochemical techniques for obtaining polymeric thin films. Thus, the polymer films obtained by plasma polymerization are pinhole-free, chemically inert, extremely uniform, thermally stable, ultrathin, and strongly adherent to a great class of different substrates [28,29]. The method does not require the use of solvents, and many steps are also eliminated, the plasma polymerization being carried out in a single step. Plasma polymerization is intensively used to obtain thin layers of conductive polymers due to the possibility of their use in various electronic devices [25–27].

Polyaniline and polyaniline-like thin films have been obtained by different plasma polymerization methods, such as capacitively and inductively coupled RF and DC-glow discharges [30–32]. Heterostructure devices based on semiconducting polymer/inorganic composite material have been fabricated using reactive magnetron sputtering and plasma polymerization in a sequential process [25]. A thin film of polyaniline was synthesized in an atmospheric pressure plasma reactor at low voltage, resulting in conductive polymer thin films with enhanced transparency [26]. Pattyn et al. used low-pressure radio-frequency (RF) plasmas to analyze the contribution of negative ions to polyaniline deposition [27].

The present study investigates the polyaniline–polydiphenylamine (PPAni-PDPA) compound realized by the plasma polymerization technique. In this work, we combine aniline and diphenylamine precursors to obtain, for the first time, polyaniline–polydiphenylamine thin films by plasma polymerization. This deposition method leads to PPAni-PDPA thin films being easily processed on substrates, with interesting semiconducting characteristics for potential use in organic electronic devices. FT-IR spectroscopic studies show PPAni, PDPA, and PPAni-PDPA compounds with similar characteristics that are produced by chemical and electrochemical syntheses. High-resolution transmission electronic microscopy (HR-TEM) showed ordered domains in the polymer thin films, which were confirmed by X-ray diffraction and selected area electron diffraction (SAED). I–V characteristics are the reasons for the semiconducting behavior of these materials.

## 2. Materials and Methods

Aniline (99%, Merck, freshly distilled, Darmstadt, Germany) and diphenylamine (≥99%, Merck, Darmstadt, Germany) were used to obtain plasma polymers from monomers and monomer blends. For the polymerization process of pure diphenylamine, a solution of diphenylamine in methanol was used (0.3g/1 mL methanol). For a good dispersion of the solid diphenylamine monomer in the liquid aniline monomer, the mixture (diphenylamine 30% wt) was placed in an ultrasonic bath for thirty minutes at 55 °C to form a liquid precursor. Both the aniline monomer and the liquid aniline/diphenylamine mixture were used for plasma polymerization without the addition of any solvents. The handling and preparation of all chemicals were done under an airflow chemical fume hood.

The DC-glow discharge plasma reactor (Figure 1) used for plasma polymerization had a Rogowski profile electrode as the anode and a hollow cylindrical cathode with Laval geometry, where the end towards the anode had a negative Rogowski shape.

If the upper end of the cathode (position 0 on the ruler from Figure 1) is taken as a reference, the substrate for deposition can be positioned in the plasma flow at different distances, positive or negative, from it. During the deposition process, the substrate is maintained at room temperature, and the substrate bias is kept at the floating potential.

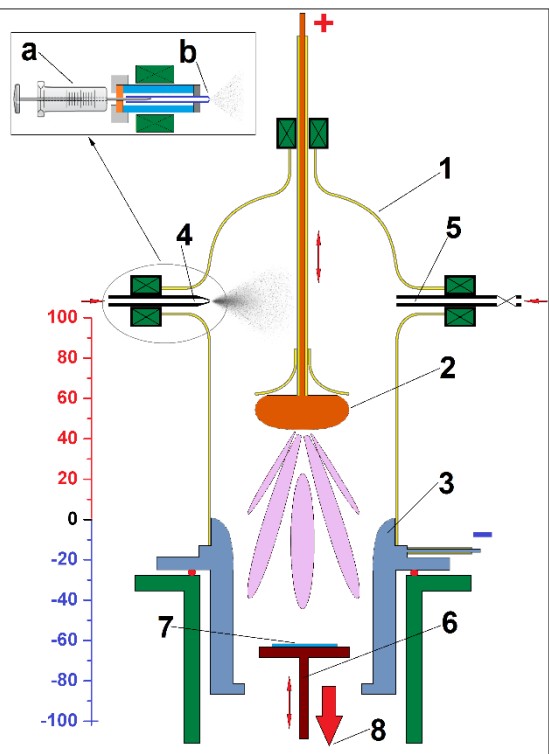

**Figure 1.** Schematic representation of the DC plasma reactor (**1**—glass reaction chamber; **2**—anode; **3**—cathode; **4**—precursor injection system; **5**—gas injection system; **6**—substrate holder; **7**—substrate (Si wafer); **8**—exhaust of chemical vacuum pump). In the inset is the precursor injection system (**a**—microliter syringe; **b**—spray nozzle).

As the distance between the anode and cathode is variable, the plasma column is easily configurable depending on the pressure in the reactor, the potential difference between the electrodes, and the monomer flow.

Several runs of plasma PPAni, PDPA, and PPAni-PDPA syntheses were done. The thin films were deposited on double-polished silicon wafers—Si (100) (n-type) substrates. For all experiments, the distance from the anode to the cathode was held at 50 mm, and the sample holder was placed at 60 mm inside the cathode. After the reaction chamber was closed, the reactor volume and substrates were flushed with a flux of argon (10 min) and then exposed to a vacuum (20 min) to allow the complete removal of trapped atmospheric gases. The precursors are introduced into the reaction chamber by an injection system with a glass microliter syringe coupled to a spray nozzle fixed in the plasma discharge chamber. This system allows both fine spraying of the precursor in the reactor as well as control of the amount of precursor introduced. Additionally, due to the variable position of the anode along the reaction chamber, the precursor can be injected into the plasma flow and above the anode. For all experiments in this work, the precursors were injected into the reaction chamber above the anode. In these conditions, the flow of precursors, which arrive in the plasma zone in a vapor state, is distributed as uniformly as possible. The rate of injection of all precursors was set at 0.0375 mL per min for 10 min in total. The plasma parameters are given in Table 1. During the plasma polymerization process, the system pressure was 1–18 Pa, measured with a Pirani gauge. After deposition, the flow of monomers and DC plasma was stopped and the system was degassed for 15 min, ensuring the complete removal of organic vapors. During the entire plasma polymerization process, the reaction chamber was pumped by a 5.9 m$^3$ h$^{-1}$ chemistry-hybrid pump (a two-stage rotary vane pump and a two-stage chemistry diaphragm pump).

**Table 1.** Process parameters for PPAni, PDPA, and PPAni-PDPA.

| Plasma Polymer | Plasma Power (W) | | Plasma Potential (V) | | Pressure (Pa) | | Deposition Time (s) | Film Thickness (nm) |
|---|---|---|---|---|---|---|---|---|
| | Initial | Final | Initial | Final | Initial | Final | | |
| PPAni | 3.5 | 7.1 | 350 | 710 | 1.33 | 18.33 | 600 | ~165 |
| PDPA | 3.5 | 5.4 | 350 | 540 | 1.33 | 13.66 | 600 | ~155 |
| PPAni-PDPA | 3.5 | 6.8 | 350 | 680 | 1.33 | 16.33 | 600 | ~160 |

Polyaniline (PPAni), polydiphenylamine (PDPA), and polyaniline–polydiphenylamine (PPAni-PDPA) thin films were characterized structurally by FT-IR spectroscopy using a Thermo-Nicolet Nexus spectrometer (Madison, WI, USA) with a resolution of 8 cm$^{-1}$. The spectra were measured in the 600–4000 cm$^{-1}$ range in absorption mode. Plasma emission spectra were obtained with an Ocean Optics HR-2000 UV–vis spectrophotometer (Orlando, FL, USA) with optical fiber at a 0.1 nm resolution. TEM, HR-TEM, and SAED images were obtained with Philips CM-120 (Eindhoven, The Netherlands) (0.4 nm resolution), and XRD patterns were recorded automatically with Shimadzu XRD-6000 (Kyoto, Japan) in a Bragg-Brentano configuration using a Ni filter and CuKα radiation (λ = 1.54056 Å).

All measurements are performed for films deposited on silicon substrates. TEM, HR-TEM, and SAED images are recorded with samples deposited on the carbon-coated copper grid. A key limitation in the TEM study for many polymer thin films is sample preparation. For the processing of TEM samples, the large difference in hardness between the silicon substrate and the deposited polymer films and the fact that the surface roughness of the polished silicon wafers was below 0.1 nm were taken into account. The samples for TEM investigation were obtained by scraping the polymer film from the silicon surface with a DiATOME ultra 35° diamond knife that was attached to a trough that was filled with distilled water. The film obtained by scraping was collected in this trough and then moved to the copper grid to be viewed under the microscope. The TEM images obtained were processed by an image processing software (version 1.37), ImageJ.

For electrical characterizations, the samples were placed in a home-built conductivity cell to investigate the dependence of current density on the voltage at room temperature. A bias voltage in the range from −4 to 4 V was applied, and the current flowing across the sample was measured by a Keithley instrument, the 2400 Series Sourcemeter model. All the measurements are carried out under a Faraday box in inert atmosphere (Ar). Data acquisition and analysis of the data were completely automated by employing the built-in software (LabTracer—version 2) of the power sourcemeter. The electrical measurements were carried out in an asymmetric configuration of Ag/plasma film/Si (100) (n-type)/Ag. This configuration is more appropriate for identifying the charge accumulated at interfaces between polymer and metal semiconductors and the transport mechanisms in plasma polyaniline (PPAni), polydiphenylamine (PDPA), and polyaniline–polydiphenylamine (PPAni-PDPA).

## 3. Results and Discussion

### 3.1. Optical Emission Spectroscopy of Plasmas

Low-resolution plasma emission spectra, in the range of 400–850 nm, for PPAni, PDPA, and PPAni-PDPA are presented in Figure 2. The spectra were collected during the plasma polymerization process for a discharge current value of 10 mA.

Optical emission lines from atomic H, H$_2$, CH, C$_2$, and Ar were observed. Argon emission lines were mainly observed in the 675–850 nm range [33]. The emission lines at 602 nm corresponded to H$_2$, which resulted from the recombination reaction of active species in plasma [34]. H atom emission lines from the Balmer series (H$_\alpha$ at 656 nm and H$_\beta$ at 486 nm) [35] could be attributed to dehydrogenation reactions or the fragmentation of C$_x$H$_y$ groups [34]. The emission line from 516 nm (transitions d$^3\Pi_g \rightarrow$ a$^3\Pi_u$ of the Swan band system) is attributed to C$_2$ radicals [36]. The emission lines of CH species at 431 nm

(transitions $A^2\Delta \rightarrow X^2\Pi$ of the Swan band system) come from the dissociation products of $CH_x$ [34].

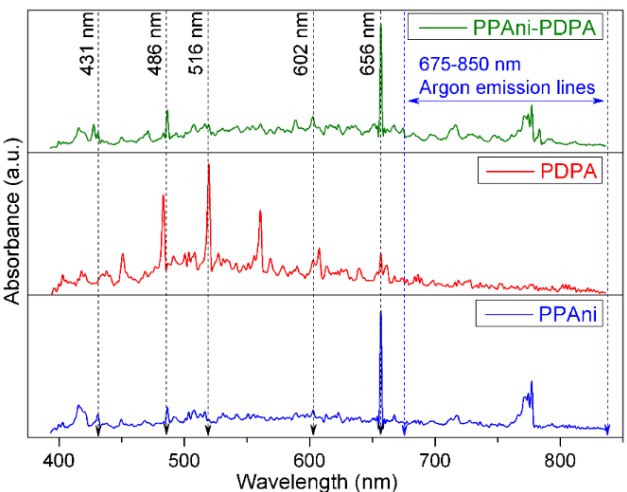

**Figure 2.** Plasma emission spectra for PPAni, PDPA, and PPAni-PDPA.

### 3.2. FT-IR Analyses

FT-IR spectroscopy studies show that PPAni (Figure 3) has an emeraldine base (PPAni-EB) [37]. The band at 827 cm$^{-1}$ in PPAni is attributed to aromatic p-substituted rings, indicating a low crosslinking in the polymer structure. The C–N groups are present in the IR spectra, with bands at 1251 and 1310 cm$^{-1}$, respectively. The bands at 1495 and 1595 cm$^{-1}$ represent the contributions from aromatic and quinoid groups, respectively. In addition, the 1441 cm$^{-1}$ band corresponding to C–C asymmetric ring vibrations gives supplemental information about enchained aromatics and not crosslinking. The bands at around 2926 and 3028 cm$^{-1}$ are typical for C–H and NH$_2$ stretching vibrations and a small amount of substitution in the ortho and meta positions of the aromatic rings. The N–H stretching band at 3361 cm$^{-1}$ is well resolved.

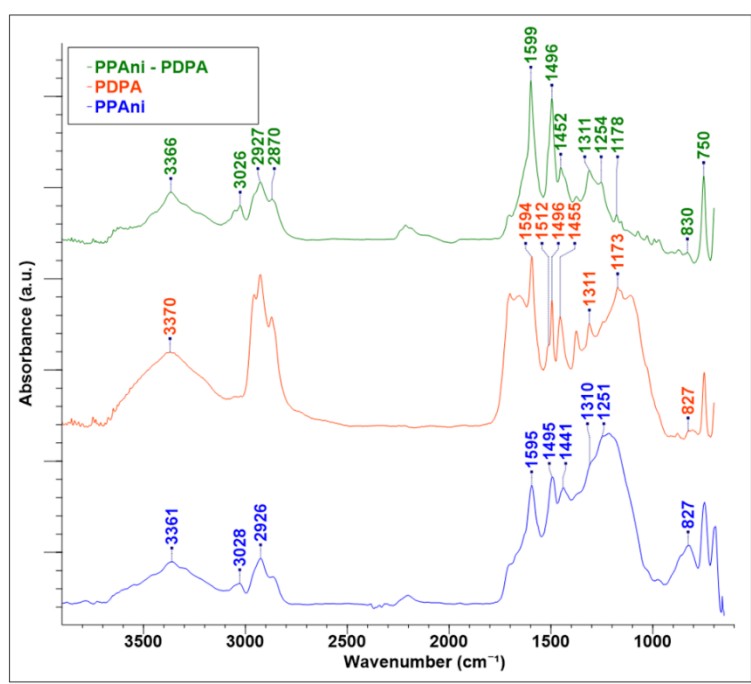

**Figure 3.** FT-IR spectra of plasma polymer thin films PPAni, PDPA, and PPAni-PDPA.

The same mode in the PDPA spectrum (Figure 3) shows that the polymer obtained by the plasma polymerization technique is very appropriate for polymers realized by chemical or electrochemical methods [22,38]. The absorption band at 827 cm$^{-1}$ corresponds to C–H out-of-plane vibrations, indicating the presence of 1,4-substituted benzene rings. This informs us that the monomer units in PDPA are bounded through C–C phenyl–phenyl coupling. The C–H bending vibrations of diphenoquinoneimine groups in PDPA are presented in a spectrum around 1173 cm$^{-1}$. The aromatic –C–N stretching band is present in the structure of PDPA (1310 cm$^{-1}$). The benzenoid and quinoid ring stretching bands are present at 1496 and 1594 cm$^{-1}$. The band at 1512 cm$^{-1}$ represents the ring stretching mode of the phenyl group's NH bending vibration of the secondary aromatic amine in PDPA. The secondary (N–H) stretching band is presented in PDPA at 3370 cm$^{-1}$.

The FT-IR spectrum of PPAni-PDPA is similar to the PPAni and PDPA spectra. This is due to the fact that PPAni and PDPA have similar frequencies in IR. On the other hand, this fact can indicate that the PPAni-PDPA thin film is a possible copolymer of PPAni and PDPA. The IR characteristic modes for PPAni, PDPA, and PPAni-PDPA, with the bibliographic references for polyaniline and polydiphenilamine obtained by chemical and electrochemical methods, are given in Table 2.

**Table 2.** FT-IR absorption frequencies for PPAni, PDPA, and PPAni-PDPA.

| Wave Number (cm$^{-1}$) | Assignment | Plasma Polymer | | |
|---|---|---|---|---|
| | | PPAni | PDPA | PAni-PDPA |
| 747 (ortho) | aromatic ring substitutions, out-of-plane (C–H) bending vibration [39,40] | 746 | 748 | 750 |
| 830 (para) | aromatic ring substitution [30,39] | 827 | 827 | 830 |
| 873, 692 | meta substitutions, 1,3 disubstitution in benzene ring [41] | 692 | 878 | 875 |
| 995, 971, 909 | C–H out-of-plane bending vibrations [41] | 973 | - | 970 |
| 1070 | quinoid ring –NH$^+$– benzoid ring stretching vibrations [40] | - | - | 1072 |
| 1173 | C–H bending vibration in quinoid ring [38,42] | 1177 | 1173 | 1178 |
| 1255 | C–N$^+$ stretching vibrations in aromatic primary amine [40] | 1251 | 1248 | 1254 |
| 1285–1315 (aromatic) 1290–1300 (quinone) | C=C, or aromatic (C–N) stretching vibrations of aromatic ring, CH bending [42,43] | 1310 | 1311 | 1311 |
| 1373 | C–H symmetric deformation vibrations in –CH$_3$ [44] | 1373 | 1376 | 1376 |
| 1450 | C=C stretching vibrations of benzenoid ring [40] | 1441 | 1455 | 1452 |
| 1498 | NH bending of aromatic secondary amines [39] | 1495 | 1496 | 1496 |
| 1505 | C=C stretching of benzene ring [38] | - | 1510 | 1512 |
| 1596, 1580–1615 | HN=quinoid=NH imine, N–H deformation vibrations of primary aromatic amine [45], quinone ring stretching [20] | 1595 | 1594 | 1599 |
| 1650 | C=N stretching vibrations of quinoid ring [40] | - | 1654 | - |
| 2862 | C–H vibrations in CH$_2$ [40] | 2862 | 2872 | 2870 |
| 2750–3000 (aromatic) | NH$_2$ stretching and C–H stretching vibrations in CH$_3$ [38,42] | 2926 | 2927 | 2927 |
| 3027 | | 3028 | - | 3026 |
| 3100–3400 (aromatic) | N–H stretching vibrations [41,42,46] | 3361 | 3370 | 3366 |

### 3.3. TEM Analyses

TEM images of the PPAni films are presented in Figure 4 at two magnifications, with scale bars of 500 nm (Figure 4a) and 100 nm (Figure 4b). Structure characterization by TEM illustrates that the PPAni films have a homogeneous structure, are organized into smooth sheets with a hierarchical arrangement, and show irregular boundaries.

TEM images for PDPA are presented in Figure 5 at three magnifications, with scale bars of 1000 nm (Figure 5a,b), 200 nm (Figure 5c), and 100 nm (Figure 5d); they point out another type of morphology, totally different from that of the PPAni polymer. The PDPA films show morphologies that highlight relatively ordered structures, similar to ultra-long

cylindrical micelles with diameters of approximately 30 nm. The organization of them in relatively ordered or aggregated structures can be observed.

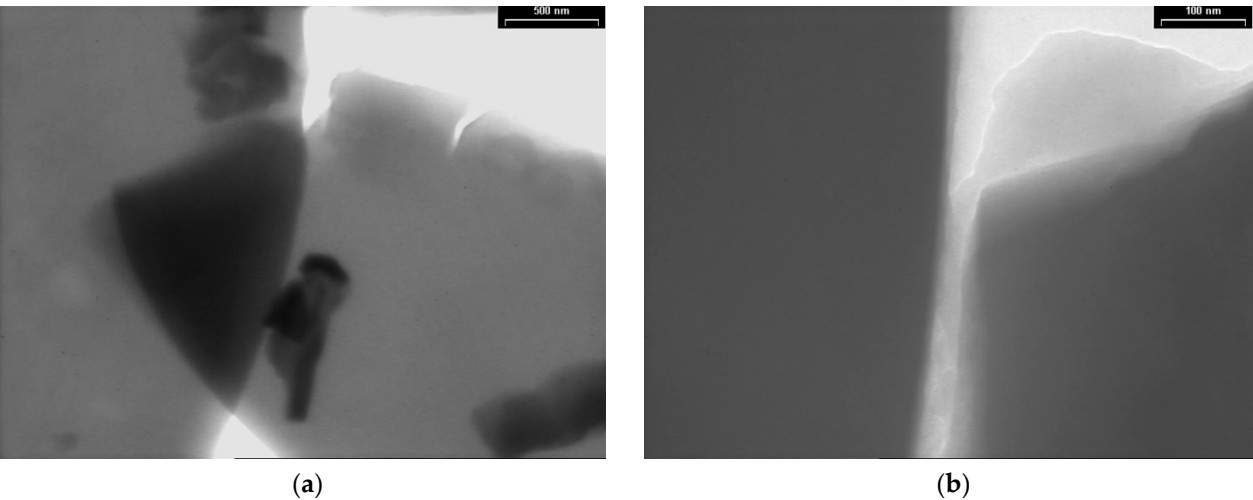

(**a**)                                    (**b**)

**Figure 4.** TEM images of PPAni thin films at various magnifications, with scale bars (**a**) 500 nm; (**b**) 100 nm.

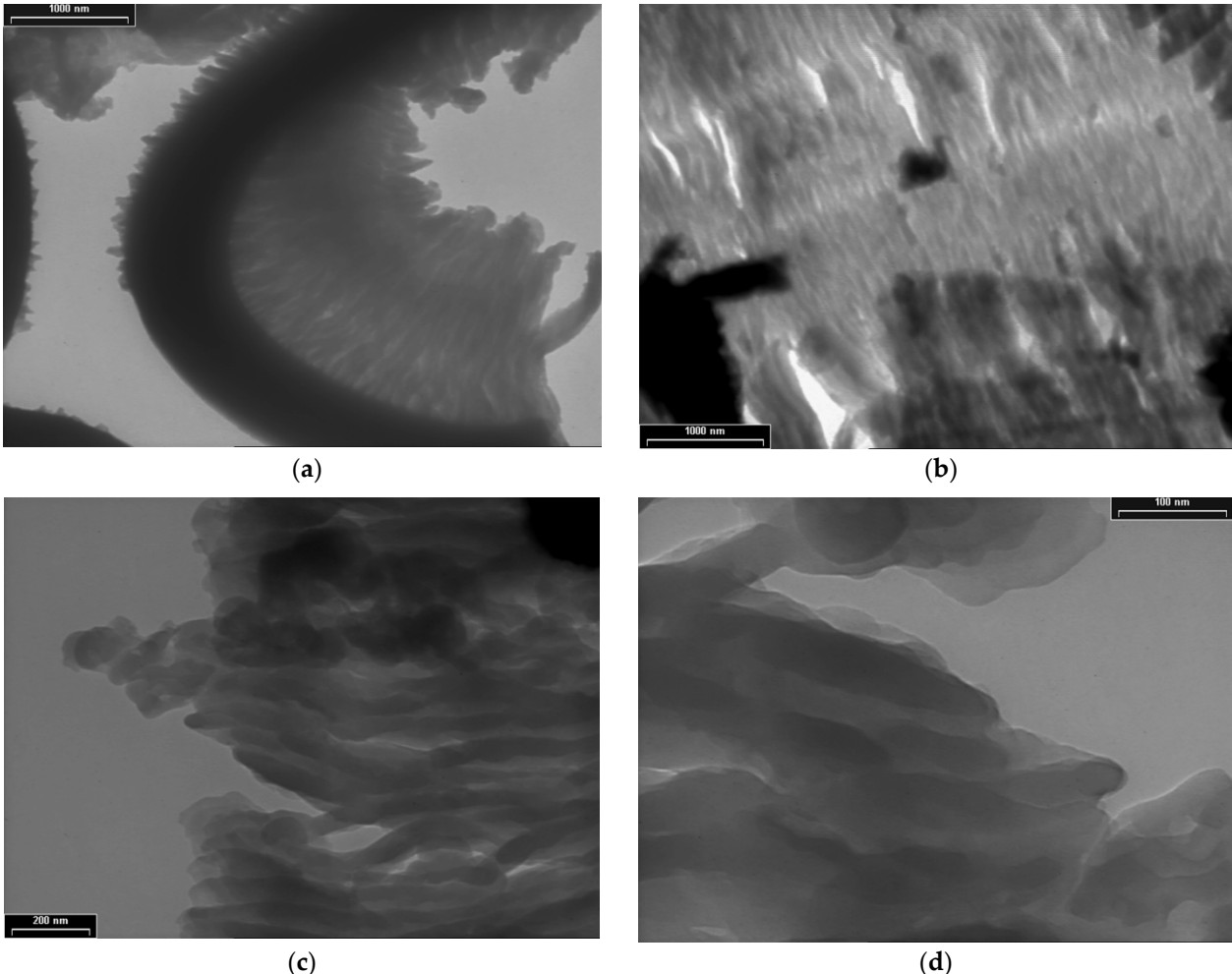

(**a**)                                    (**b**)

(**c**)                                    (**d**)

**Figure 5.** TEM images of PDPA thin films at various magnifications, with scale bars (**a**) and (**b**) 1000 nm; (**c**) 200 nm; (**d**) 100 nm.

Figure 6 shows the TEM images of PPAni-PDPA films at three magnifications, with scale bars of 1000 nm (Figure 6a,b), 200 nm (Figure 6c), and 100 nm (Figure 6d); the figures reveal the presence of highly ordered tubular morphology. Structures with a cylindrical aspect are aligned along an arbitrary direction, are relatively long, and have diameters of around 50 nm. Copolymerization of aniline with diphenylamine by plasma polymerization leads to an improvement in the orientation observed in polydiphenylamine films. Additionally, the presence of cylindrical and ordered structures in PPAni-PDPA films suggests the possibility of forming structures similar to highly oriented block copolymer systems.

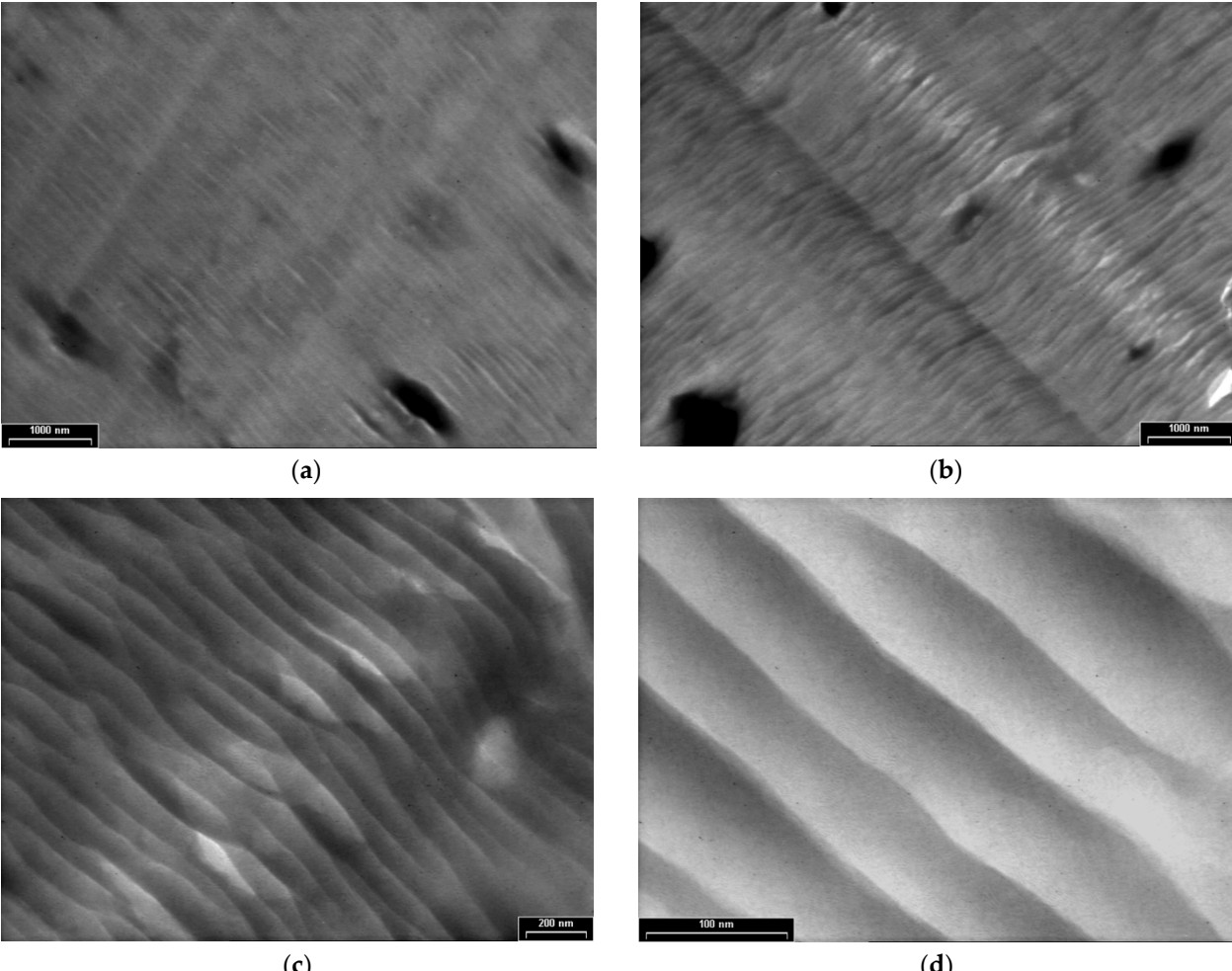

**Figure 6.** TEM images of PPAni-PDPA thin films at various magnifications, with scale bars (**a**) and (**b**) 1000 nm (**c**) 200 nm (**d**) 100 nm.

### 3.4. XRD, SAED and HR-TEM Analyses

In Figure 7, X-ray diffractograms of PPAni, PDPA, and PPAni-PDPA films are presented. It can be observed that most of the diffraction lines are common to the three structures. In the case of the PPAni-PDPA structure, a distinct line appears, corresponding to an interchain distance of 0.756 nm and a size of the ordered domains of 56.39 nm. Additionally, the line corresponding to an interchain distance of around 0.135 nm shows three well-defined splits.

The average crystallite size for the polymer structures, also called the extension of the ordered domains *t*, was deduced from the width of the diffraction lines using Scherrer's relation, Equation (1) [47]:

$$t = \frac{0.93\ \lambda}{w\ \cos\theta_B} \tag{1}$$

where $t$ is the average size of the crystallites, $\lambda$ is the wavelength of X radiation, $\theta_B$ is the Bragg angle, and $w$ is the measurement of the peak full width at half maximum.

The values of the average crystallite size calculated with Equation (1) for PPAni, PDPA, and PPAni-PDPA films are presented in Table 3. It can be observed that the largest expansion of the ordered domains is found in the PPAni-PDPA structure.

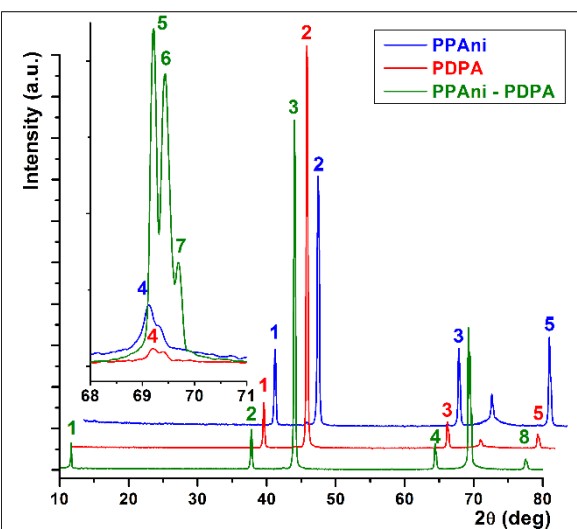

**Figure 7.** XRD patterns of PPAni, PDPA, and PPAni-PDPA.

**Table 3.** Indexing of diffraction lines for PPAni, PDPA, and PPAni-PDPA.

| Plasma Polymer | Diffraction Line | $2\theta$ | $d$ (nm) | $t$ (nm) |
|---|---|---|---|---|
| PPAni | 1 | 37.74 | 0.238 | 37.72 |
| | 2 | 43.98 | 0.205 | 35.29 |
| | 3 | 64.34 | 0.144 | 34.36 |
| | 4 | 69.12 | 0.135 | 35.32 |
| | 5 | 77.46 | 0.123 | 33.56 |
| PDPA | 1 | 37.86 | 0.237 | 34.59 |
| | 2 | 44.08 | 0.205 | 36.84 |
| | 3 | 64.42 | 0.144 | 34.38 |
| | 4 | 69.20 | 0.136 | 47.71 |
| | 5 | 77.54 | 0.122 | 29.62 |
| PPAni-PDPA | 1 | 11.68 | 0.756 | 56.39 |
| | 2 | 37.80 | 0.237 | 34.59 |
| | 3 | 44.02 | 0.205 | 35.29 |
| | 4 | 64.38 | 0.144 | 34.37 |
| | 5 | 69.22 | 0.135 | 73.40 |
| | 6 | 69.44 | 0.135 | 50.29 |
| | 7 | 69.70 | 0.134 | 47.85 |
| | 8 | 77.48 | 0.123 | 29.61 |

SAED and HR-TEM studies for PPAni (Figure 8), PDPA (Figure 9), and PPAni-PDPA (Figure 10) highlight the tendency of the obtained thin films to form well-organized domains. The ring diffractograms are typical for polycrystalline materials, the radius of the rings being inversely proportional to the interplanar spacings of the lattice planes of the crystals.

The HR-TEM and SAED images confirm the X-ray diffraction studies. The interplanar spacings calculated from the SAED and HR-TEM images show a good correlation with the results obtained from XRD.

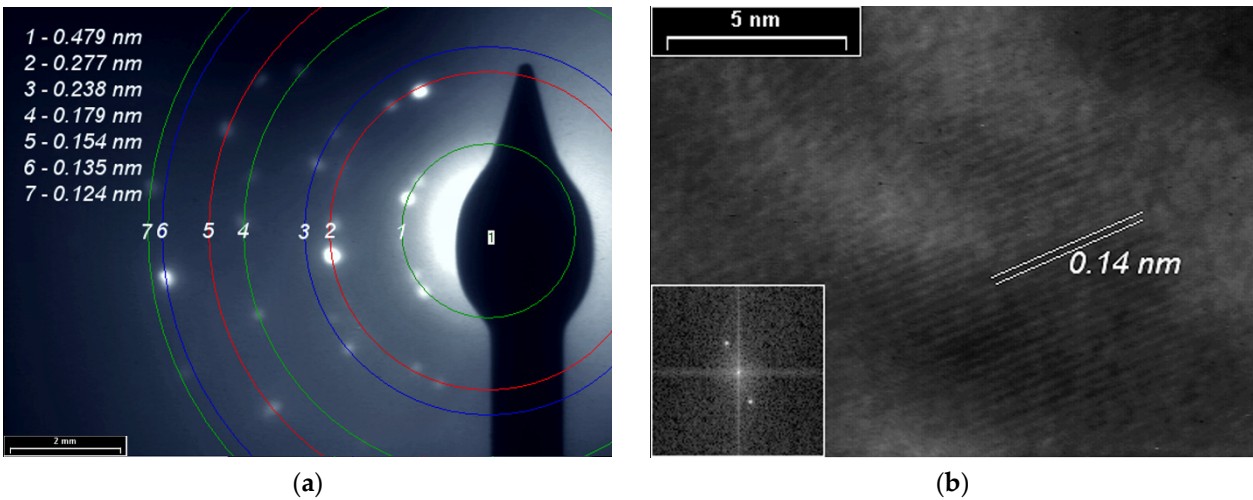

**Figure 8.** (**a**) SAED pattern; (**b**) HR-TEM images and in inset the fast Fourier transform (FFT) pattern of the PPAni film.

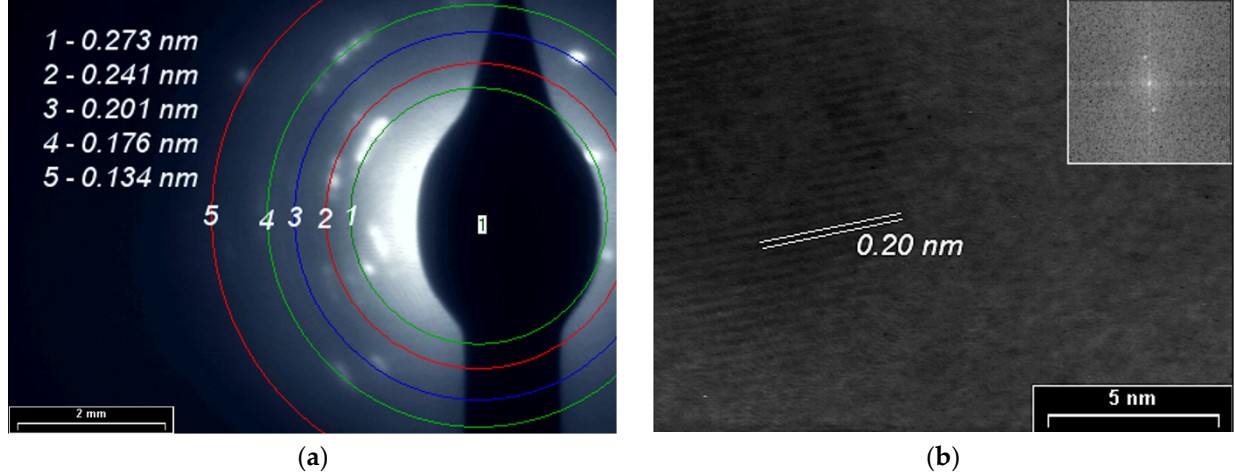

**Figure 9.** (**a**) SAED pattern; (**b**) HR-TEM images and in inset the fast Fourier transform (FFT) pattern of the PDPA film.

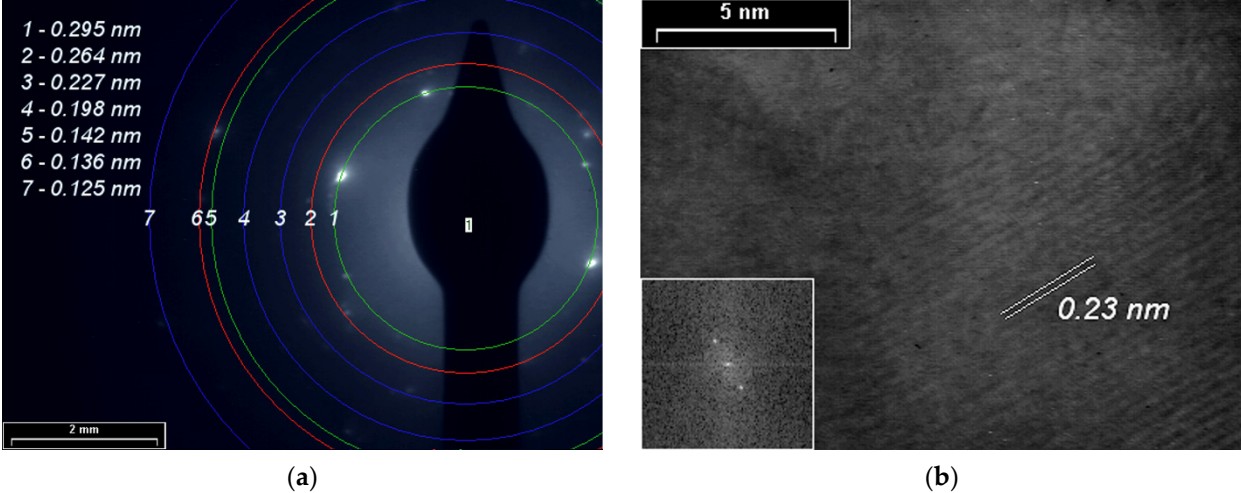

**Figure 10.** (**a**) SAED pattern; (**b**) HR-TEM images and in inset the fast Fourier transform (FFT) pattern of the PPAni-PDPA film.

In this study, a non-uniform intensity distribution along the ring's circumference, where the ordered domains are interspersed in amorphous areas, was observed for all three SAED images; this shows that the thin films grown by plasma polymerization are textured materials.

### 3.5. I–V Characteristics

I–V characteristics in an asymmetric configuration for Ag/plasma film/Si(100)(n-type)/Ag were studied for PPAni, PDPA, and PPAni-PDPA thin films. This kind of investigation is similar to electrical spectroscopy, which allows us to establish the conduction mechanisms of the carrier transport. Figure 11 shows the I–V characteristics for PPAni, PDPA, and PPAni-PDPA, from which it is evident that the PPAni-PDPA structure shows a different behavior than that of the PPAni and PDPA structures. If the PPAni and PDPA structures have relatively symmetrical characteristics that are specific to an intrinsic semiconductor, the PPAni-PDPA structure has an asymmetric behavior.

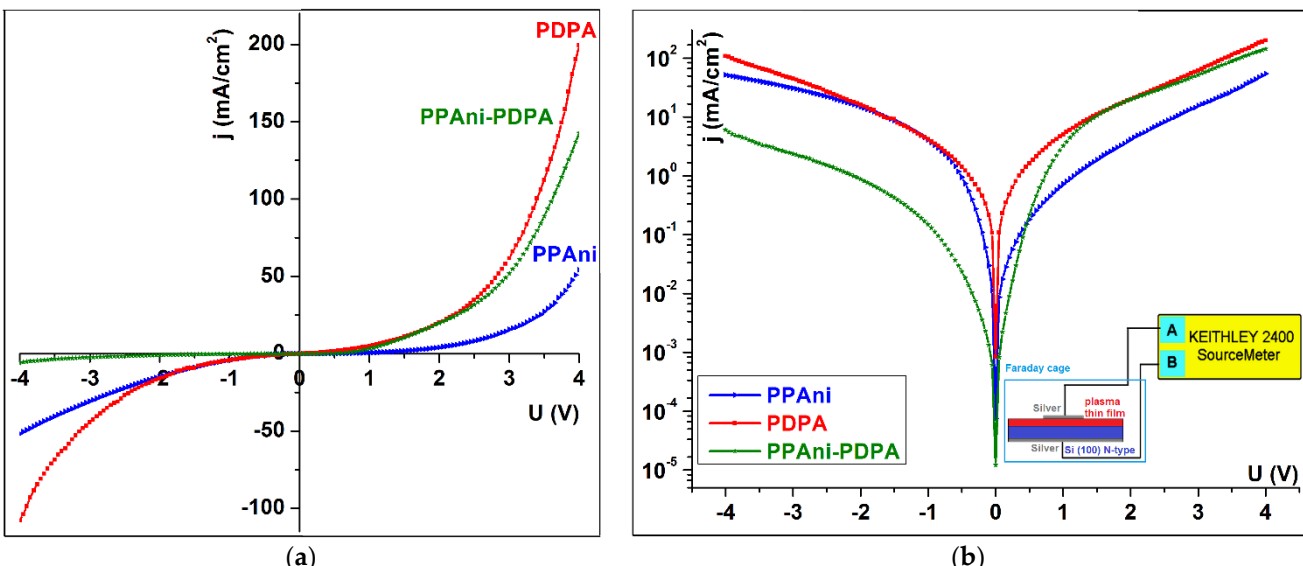

**Figure 11.** The plot of the current–voltage of the PPAni, PDPA, and PPAni-PDPA (**a**) linear plot; (**b**) semi-log plot.

To establish the possibility of the appearance of the Schottky barrier effect for these configurations, the graphs j vs. $U^{1/2}$ (Figure 12a) and lnj vs. $U^{1/2}$ (Figure 12b) were plotted. From the characteristics, it was highlighted that only the PPAni and PDPA structures present a lnj vs. lnU dependence close to the linear behavior characteristic of Schottky behavior. The PPAni-PDPA structure presents two distinct regions, depending onlnj vs. lnU, that present characteristics close to a linear behavior with a large difference between the slopes of lnj vs. lnU in the two regions; this fact indicates a weak Schottky behavior.

The lnj vs. lnU graph depicted in Figure 13 shows the presence of multiple conduction mechanisms in the three structures. The conduction mechanism encountered in the PDPA structure is similar to that identified in the PPAni structure and presents a continuous transition from an ohmic region ($j \sim U^{1.0}$, $j \sim U^{1.1}$) to a transport governed by relatively shallow traps, where the space charge limited current is dominant ($j \sim U^{1.7}$, $j \sim U^{1.8}$) in a trap-filled limited conduction mechanism ($j \sim U^{3.5}$, $j \sim U^{3.2}$).

The plot lnj vs. lnU of the PPAni-PDPA structure shows three regions, depending on the bias voltage. The first is described by the space charge limited current (SCLC) with shallow traps ($j \sim U^{1.6}$). The second is given by the trap filling limited current (TFLC), where $j \sim U^{3.9}$, and the last is an SCLC regime where all the traps are filled ($j \sim U^{2.5}$) [48].

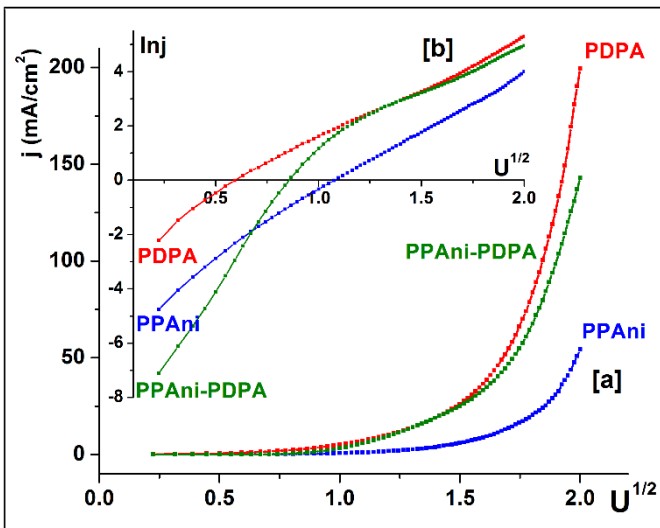

**Figure 12.** (**a**) Plots of j vs. $U^{1/2}$ and (**b**) identification of Schottky behavior for PPAni, PDPA, and PPAni-PDPA films.

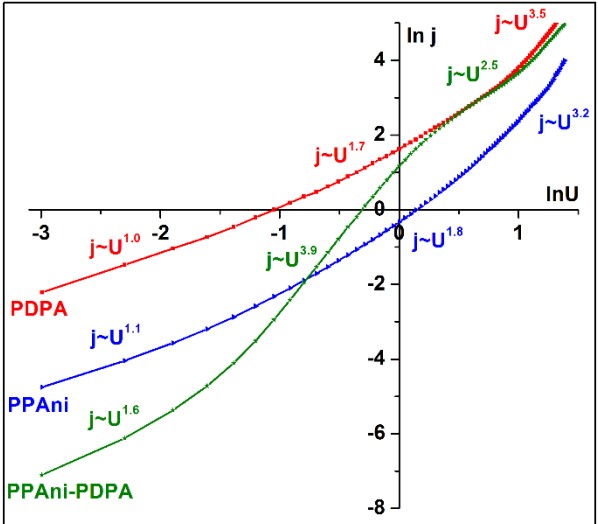

**Figure 13.** lnj vs. lnU plots for PPAni, PDPA, and PPAni-PDPA films.

## 4. Conclusions

In this paper, we show a contribution to the organic semiconducting field. We demonstrate that plasma polymerization may be a good competitor process to chemical and electrochemical syntheses for semiconducting polymers; in this way, easily processable polymers can be obtained.

FT-IR spectroscopic studies show PPAni, PDPA, and PPAni-PDPA compounds with similar characteristics to polymers that are produced by chemical and electrochemical syntheses. The HR-TEM images of PPAni and PDPA present organized structures such as crystalline planes; this was confirmed by XRD studies and SAED. The interplanar distances varied between 0.75 to 0.12 nm. Besides their organization at short-distance (molecular level), the thin films of PPAni-PDPA present an organization at long-distance as fascicles. The current–voltage characteristic shows three different conductivity regions: ohmic, space charge limited current (SCLC), and trap filling limited current (TFLC).

**Author Contributions:** Conceptualization, F.N. and C.N.; methodology, F.N. and C.N.; validation, F.N. and C.N.; formal analysis, F.N. and C.N.; investigation, F.N., C.N. and G.P.; data curation, F.N. and G.P.; writing—original draft preparation, C.N.; writing—review and editing, F.N., C.N. and G.P. All authors have read and agreed to the published version of the manuscript.

**Funding:** Publication of this work was funded by The Ministry for Research, Innovation and Digitization, Core Program, grant number: PN 14N/2019 (MICRO-NANO-SIS PLUS).

**Institutional Review Board Statement:** Not applicable.

**Informed Consent Statement:** Not applicable.

**Data Availability Statement:** Data sharing is not applicable to this article.

**Conflicts of Interest:** The authors declare no conflict of interest.

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
