# Peer review of "Plasma-Polymerized Aniline–Diphenylamine Thin Film Semiconductors"

_coatings, doi:10.3390/coatings12101441_

Round 1
Reviewer 1 Report
This manuscript reports on the plasma polymerized aniline – diphenylamine thin film semiconductors. Certainly, the growing by DC plasma polymerized technique of semiconducting polymer thin films using a mixtures of aniline-diphenylamine as precursor.
This research direction is dynamical developing and, of course, is timely and challenging for various areas of modern chemistry and physics. The authors used the traditional toolkit of physical-chemical methods in their investigations.
However, before publication, the following issues should be addresses:
1. How were the manipulations with the initial monomers carried out: in argon or in air? Was any solvent used for the work with pure diphenylamine (version without aniline)? How was diphenylamine purified given that distillation is not usually applied to solids?
2. The experiments on preparation of samples for physical-chemical study are not described in sufficient detail. For example, it remains unclear how samples were prepared for TEM and HR-TEM.
3. On page 4 it is stated that «The emission line from 516 nm (transitions d3Πg →a3Πu of the Swan band system) is attribute to C2 radicals [35].» Have similar structures been studied in detail, for example, by EPR spectroscopy?
4. Can the amount of free radicals captured by the target films be related to the spin mobility? Is it possible to control the potential electrical properties (conductivity) through the regulating of spin mobility?
5. Is it possible to investigate the conduction mechanism with the help of EPR?
Author Response
Thank you for your suggestions and comments on our manuscript.
Below are our answers to your comments:
- How were the manipulations with the initial monomers carried out: in argon or in air? Was any solvent used for the work with pure diphenylamine (version without aniline)? How was diphenylamine purified given that distillation is not usually applied to solids?
Response: “Material and Methods” section was improved by new detail about experiments. Changes are in red.
- The experiments on preparation of samples for physical-chemical study are not described in sufficient detail. For example, it remains unclear how samples were prepared for TEM and HR-TEM.
Response: “Material and Methods” section was improved by new detail about preparation of samples. Changes are in red.
- On page 4 it is stated that «The emission line from 516 nm (transitions d3Πg→a3Πuof the Swan band system) is attribute to C2 radicals [35].» Have similar structures been studied in detail, for example, by EPR spectroscopy?
Response: At a first search in the specialized literature, there are no studies in which EPR spectroscopy is used for polyaniline or polydiphenylamine obtained by plasma polymerization. However, there are studies in which EPR spectroscopy is used for polyaniline obtained by chemical synthesis [1], or polydiphenylamine obtained by electrochemical synthesis [2].
[1] Y.O.Mezhuev et all., EPR monitoring of aniline polymerization: Kinetics and reaction mechanism, Synthetic Metals, 280, (2021), 116871
[2] S. –J. Dong et all., Electrochemical synthesis and characterization of polydiphenylamine. Chinese Journal of Chemistry, 10(1), (1992), 10–16
- Can the amount of free radicals captured by the target films be related to the spin mobility? Is it possible to control the potential electrical properties (conductivity) through the regulating of spin mobility?
Response: Electron paramagnetic resonance (EPR) is the technique used with excellence to detect and characterize free radicals in polymer films. In these conditions, the amount of free radicals captured in the polymer films can be related to the spin mobility [1].
[1] P. Silva et all., Use of electron paramagnetic resonance to evaluate the behavior of free radicals in irradiated polyolefins. Rev. LatinAm. Metal. Mater. 28(2), (2008), 79-90
Regarding the control of electrical properties by regulating the spin mobility, there are no studies, in the specialized literature, for polymers obtained by plasma polymerization. However, there are studies in which these aspects are discussed [1, 2].
[1] V. Zayets, Spin and charge transport in materials with spin-dependent conductivity, Phys. Rev. B, 86(17), 2012, 174415
[2] C. Lefter et all., Charge Transport and Electrical Properties of Spin Crossover Materials: Towards Nanoelectronic and Spintronic Devices, Magnetochemistry, 2(18), 2016
- Is it possible to investigate the conduction mechanism with the help of EPR?
Response: We do not have experience in EPR spectroscopy, but EPR spectroscopy is a useful tool in the study of conduction mechanisms in polymers [1].
[1] S.K. Gupta et all., Electrical transport and EPR investigations: A comparative study for d.c. conduction mechanism in monovalent and multivalent ions doped polyaniline, R. Bull Mater Sci, 35, (2012), 787–794
Reviewer 2 Report
The manuscript entitled “Plasma polymerized aniline–diphenylamine thin film semiconductors” reports morphological and structural studies of the obtained thin films and its conduction mechanism.
In my opinion the manuscript could be suitable for publication in Coating journal after a minor revision to increase the reader’s interest and the accuracy of the research.
The following concerns should be addressed:
1) In the introduction session, the novelty and significance of the work should be emphasised. The advantages of plasma polymerization should be more highlighted. In addition, the potential impact of the research and why it is important, compared to other research in this field or previous studies, should be discussed.
2) What is the meaning of FTO and PTA acronyms in line 46-47?
3) Please pay attention to Figure 2 caption.
4) The Authors state that the PPAni, PDPA and PPAni-PDPA compounds have characteristics similar to those produced by chemical and electrochemical synthesis. FTIR spectra of the compounds produced by chemical and electrochemical synthesis should be added to demonstrate this theory. Alternatively, the Authors could add appropriate bibliographic references.
5) TEM analyses should be discussed in more detail.
Author Response
Thank you for your suggestions and comments on our manuscript.
Below are our answers to your comments:
1) In the introduction session, the novelty and significance of the work should be emphasised. The advantages of plasma polymerization should be more highlighted. In addition, the potential impact of the research and why it is important, compared to other research in this field or previous studies, should be discussed.
Response: The Introduction was improved. The advantages of plasma polymerization are more evident now. Also, the novelty of our research was highlighted. The new bibliographic reference was introduced. Changes are in red.
2) What is the meaning of FTO and PTA acronyms in line 46-47?
Response: We replaced “FTO/PDPA-PAni/PTA” with “Fluorine-doped tin oxide/ polydiphenylamine-polyaniline /phosphotungstic acid”. Changes are in red.
3) Please pay attention to Figure 2 caption.
Response: We modified the caption of Figure 2
4) The Authors state that the PPAni, PDPA and PPAni-PDPA compounds have characteristics similar to those produced by chemical and electrochemical synthesis. FTIR spectra of the compounds produced by chemical and electrochemical synthesis should be added to demonstrate this theory. Alternatively, the Authors could add appropriate bibliographic references.
Response: We introduced in the text supplementary references with FTIR spectra of polyaniline and polydiphenilamine compounds produced by chemical and electrochemical. Also, in text we specified that the bibliographic references of IR characteristic modes for polyaniline and polydiphenilamine obtained by chemical and electrochemical methods are given Table 2.
5) TEM analyses should be discussed in more detail.
Response: The section with TEM analyses was improved. The changes are in red.